Alternative reproductive strategies in black-winged territorial males of Paraphlebia zoe (Odonata, Thaumatoneuridae)

Rivas-Torres Anais anarivas@uvigo.es arivasto@gmail.com 1
Sánchez-Guillén Rosa Ana 2
Cordero-Rivera Adolfo 1
1 ECOEVO Lab, Departamento de Ecoloxía e Bioloxía Animal, Universidade de Vigo , Pontevedra , Galiza , Spain
2 Department of Ecology and Animal Biology, INECOL , Xalapa , Veracruz , Mexico
Hedrick Ann
Electronic publication date: 2019 Feb 20
Publication date: 2019
Volume: 7
Electronic Location ID: e6489
Received 2018 Sep 7; Accepted 2019 Jan 19
Copyright: ©2019 Rivas-Torres et al.
Copyright year: 2019
Copyright holder: Rivas-Torres et al.
License: This is an open access article distributed under the terms of the Creative Commons Attribution License, which permits unrestricted use, distribution, reproduction and adaptation in any medium and for any purpose provided that it is properly attributed. For attribution, the original author(s), title, publication source (PeerJ) and either DOI or URL of the article must be cited.
License URL: https://creativecommons.org/licenses/by/4.0/

Keywords: Alternative tactics, Dominance, Territoriality, Submission, Wing-spot size, Damselfly

Funding: Spanish Ministry of Economy and Competitiveness MINECO, BES-2015-071965 MINECO FEDER CGL2014-53140-P CONACYT 282922 Anais Rivas Torres, was supported by an FPI grant of the Spanish Ministry of Economy and Competitiveness (MINECO, BES-2015-071965). Funding was provided by a grant from MINECO, including FEDER funds to Adolfo Cordero-Rivera (CGL2014-53140-P) and by CONACYT (Ciencia Básica project number 282922) to Rosa Ana Sánchez Guillén. The funders had no role in study design, data collection and analysis, decision to publish, or preparation of the manuscript.

==============================
Alternative reproductive strategies are commonly associated with male dimorphism. In Paraphlebia zoe, a species of damselfly whose males are dimorphic in wing coloration, black-and-white-winged (BW) males defend territories, while hyaline-winged (HW) males usually play the role of satellites. We found that several BW males can sometimes share a territory, and we hypothesized that within this morph there are two alternative tactics: submissive and dominant. We conducted an experiment to test whether dominant and submissive roles are plastic or stable and fixed on each individual. To this end, we manipulated black and white spots of BW males in four treatments: (i) painting over white and black spots without changing their size, (ii) erasing the white spot using black painting, (iii) increasing the black spot and moving the white spot maintaining its size and (iv) control males. Additionally, we investigated the correlation between some phenotypic variables (wing asymmetry, survival and recapture probabilities) and male behaviour (in terms of quality of the territory). We found that the two behavioural roles (submissive and dominant) were not affected by the manipulative experiments, therefore suggesting that they are stable and fixed. Additionally, we found a positive correlation between body size and survival in both sexes, and a positive effect of territory quality and lifespan on mating success. Moreover, the largest and youngest BW males were the most symmetrical. We conclude that Paraphlebia zoe holds high behavioural diversity, with two types of strategies in BW males, dominant and submissive. The occurrence of this intra-morph behavioural diversity might depend on demographic factors such as population density and/or the relative frequency of the different morphs.

Introduction

The existence of alternative reproductive behaviours in individuals of the same sex is widespread in nature. Most of examples of alternative reproductive phenotypes are readily interpreted as alternative tactics within a conditional strategy, i.e., monomorphic individuals adopting one strategy or another, based on a condition-dependent trait, such as body size, fat reserves and so on (Gross, 1996). In contrast, alternative strategies linked to genetic polymorphisms are uncommon, and they are usually associated with male polymorphism in insects and other invertebrates. Examples include wing dimorphism in odonates, and size in horned beetles and cuttlefish (Nomakuchi, 1992; Emlen, 1997; Norman, Finn & Tregenza, 1999). In polymorphic species, each morph is usually associated with different behaviours, like two morphs of newts which differed in boldness (Winandy & Denoël, 2015). In harvestmen, a case of trimorphism in chelicerae size has been recently reported, associated with alternative reproductive tactics (Painting et al., 2015).

In odonates, colour polymorphism is common in females (Fincke et al., 2005; Van Gossum, Sherratt & Cordero-Rivera, 2008; Sánchez-Guillén et al., 2017). Female-limited colour polymorphism (thorax colouration) is associated with alternative reproductive strategies, normally one female has a male-like coloration and behaviour (androchrome; behaving more aggressively than the gynochrome morphs). However, male-limited colour polymorphism (wing colouration) is a rare phenomenon in damselflies and only found in some families like Calopterygidae (genus Mnais; Siva-Jothy & Tsubaki, 1989; Plaistow & Tsubaki, 2000, Platystictidae González Soriano, Novelo Gutiérrez & Verdugo Garza, 1982), Synlestidae (Samways, 2006) and in the genera Thaumatoneura and Paraphlebia, (Garrison, Von Ellenrieder & Louton, 2010; Paulson, 2004). In Diastatops obscura (Libellulidae), there are three types of males: territorial males near the riverbank (the best territories), territorial males in the middle of the river, and satellites in the riverbank vegetation, who do not defend territories (Bañuelos Irusta & Araújo, 2007). Territories can be even shared by several males, like in Libellula luctuosa, and are surrounded by non-territorial satellite males (Moore, 1987).

The damselfly Paraphlebia zoe Selys in Hagen, 1861, is a polymorphic species, with two male colour morphs, one with coloured wings, with a black spot and a white stripe (BW), while the other morph shows hyaline wings (HW), similar to females. In Megaloprepus caerulatus, a monomorphic species whose males have a similar wing pattern to BW P.  zoe males, it has been found that white wing bands of males are correlated with territory residence and they might be a signal of male condition, because they are based on wax (Schultz & Fincke, 2009; Xu & Fincke, 2015). In P. zoe, each morph has been associated with a specific reproductive strategy. BW males are more territorial and aggressive than HW males, secure more matings and copulate for shorter times (Romo-Beltrán, Macías-Ordóñez & Córdoba-Aguilar, 2009). This species is therefore a typical example of genetic polymorphism associated with alternative reproductive strategies, presumably maintained by frequency-dependent selection (Gross, 1996). Some BW males that do not defend a territory have been previously found in P. zoe (Romo-Beltrán, Macías-Ordóñez & Córdoba-Aguilar, 2009), and are considered as satellite males. Our preliminary observations suggested that BW territorial males can behave as dominants, particularly when their territory is close to the water or as submissive, when they occupy a territory away from the water, in a similar way as the behaviour described for D. obscura (Bañuelos Irusta & Araújo, 2007). Dominant BW males are highly territorial and highly aggressive against submissive males, probably giving them resource advantages (Baker, 1983), and submissive males are less aggressive, usually avoiding confrontation with dominants, but nevertheless defend their territory against satellites. Paraphlebia zoe does not show precopulatory courtship (Wong-Muñoz et al., 2013), consequently, females could be choosing territory quality instead of male quality.

Previous studies have indicated that longevity and body size are among the main correlates of male reproductive success in damselflies (e.g.,  Banks & Thompson, 1987; Cordero, 1995). The main aim of our study was to determine whether dominant and submissive roles of BW males are related to phenotypic characters, in particular body size, wing asymmetry and size of the wing pigmentation (black spot and a white stripe), and how this behavioural polymorphism relates to reproductive success. To this end, we first, using mark-recapture methods, estimated survivorship rates and the effect of body size on survival of BW males (dominant and submissive). Second, fluctuating asymmetry (Forbes, Leung & Schalk, 1997; Beck & Pruett-Jones, 2002) has been widely used as a correlate of genetic quality, because it is presumed that, for traits with bilateral expression, small deviations between sides are due to the inability of the organism to buffer environmental stressors (Leung & Forbes, 1997a). In odonates, some contrasting results about the effect of asymmetry on reproductive success have been found. Harvey & Walsh (1993) found a negative effect, while others did not find any effect (Leung & Forbes, 1997b; Koga et al., 1998; Carchini et al., 2001). We therefore analysed the effect of male wing spot asymmetry on mating success of BW males (taking into account territory quality and male strategy). Third, some studies have suggested that male mating success might be more related to the ability to be in the right place at the right time, which is expected to be related with the capacity to maintain the territory (Switzer, 2002). For this reason, we studied mating behaviour and evaluated which factors affect copulation duration, including male phenotype, time of day and temperature (e.g.,  Cordero, 1990), because these variables might affect territory tenancy and hence reproductive success. Finally, we designed a manipulative experiment to test whether dominant and submissive BW male roles are plastic or fixed over time. Based on previous studies with odonates that have found that more pigmented males have higher success (e.g., Córdoba-Aguilar, 2002) we hypothesized that increasing the black-wing spot of the BW submissive males would change their status towards dominance. This prediction is based on the idea that males adapt their aggressiveness to the result of previous interactions, so that other males would tend to become submissive in confrontations with a male with an enlarged black spot, and the manipulated male would increase its aggressiveness (Senar & Camerino, 1998). However, increasing the black spot size of the BW dominant males would not affect their status, because they are already dominant. For the white spot, there is no information about its function in this species. Therefore, if we conceal the white spot, we could either expect that BW dominant males change their status to submissive if this spot is related to aggressiveness (Schultz & Fincke, 2009; Xu & Fincke, 2015), or that they maintain the same dominance, if the spot is not related to dominance. In the case of submissive BW males, concealing their white spot would not affect their behaviour under the assumption that the white spot is not related to dominance.

Methods

Morphometrics and asymmetry

Paraphlebia zoe was studied in Teocelo, Veracruz, Mexico (19°40′42″N, 96°98′28″W) in a creek depression in a runoff water habitat, from 11 June to 25 July, 2017. Fieldwork was carried out from 9:00 to 16:30, every day if the weather allowed it (i.e., if there were no major showers), by two observers who recorded date and time of capture, male morph and body size, measured to the nearest 0.1 mm using a caliper. Males and females were captured with an entomological net and marked with a small number on the left anterior wing, using a black Staedleter permanent lumocolor pen, a method that does not affect the survival of the damselflies (Munguía-Steyer, Córdoba-Aguilar & Romo-Beltrán, 2010). The number was situated far from the wing spots (see Fig. 1), to minimize the possible effect of this mark on behaviour.

Figure 1 The four manipulative experiments.

(A) Control 1; painting over white and black spots. (B) Control 2; unmanipulated males. (C) IBS; increasing the black spot and moving the white spot maintaining its size. (D) EWS; erasing the white spot using black painting.

During our fieldwork period, HW males were extremely rare or absent, and were therefore not included in the study. For BW males, behavioural status was divided into two categories: territorial males, defined as those individuals who stayed in the same place for more than two days and returned to the same perch site after participating in a fight in flight with a conspecific male, while individuals who did not meet this criterion were considered as non-territorial BW (satellite) males (Romo-Beltrán, Macías-Ordóñez & Córdoba-Aguilar, 2009). Within the BW territorial males, we found two subcategories: the dominant territorial males and the submissive males. The subcategories of the BW males were assigned observing them during 3 days, using 10-min focal observations each day. The male which won more than 60% of the interactions in which he was involved, was labelled as the dominant, whereas the submissive male was the one that won less than 50% of the interactions (Fig. 2). Moreover, the histogram of Fig. 2 shows that there is no overlap between dominant and submissive males, and therefore they can be unambiguously identified. When a focal individual was observed perching outside the area of territories, it was considered to have switched to submissive. However, for focal males not observed during a day, we maintained their originally assigned status because they could be feeding, or making a break in their reproductive activities.

Figure 2 Histogram of the proportion of interactions won in BW males of P. zoe.

We considered dominant males, these that won more than 60% of the interactions, and submissive males these that won less than 50% of the interactions. In black, dominant BW males. In grey, submissive BW males.

Body size was measured as length from the head to the end of the abdomen (including cerci) to the nearest 0.1 mm with a caliper. After that, each BW male was photographed with the wings spread for asymmetry measurements. The software Image J (Schneider, Rasband & Eliceiri, 2012) was used to measure area of the black and white spots from BW males as proxies of spot size. To control for trait size, asymmetry was calculated as the absolute value of the difference between right and left sides, divided by the average value of both sides (Cuervo, 2000). Four estimates of asymmetry were obtained for each BW male (one for each black and white spot), and we calculated the total asymmetry by adding up the four partial estimates. To distinguish measurement error from actual asymmetry, each individual was measured twice and the variance due to replicate and wing size was estimated following Crawley (2007). A sample of ten males, marked and photographed as young individuals and recaptured after 15 to 37 days, were photographed again to study ontogenetic changes in spot size. If spot size is related to genetic quality, and is not condition-dependent, we expect spot size to remain invariable over time. This could not be compared among dominant and submissive roles because the number of young individuals that were recaptured was too low.

Estimation of survival rates

To estimate survival (Phi) and recapture (p) probabilities we analysed the recapture histories of marked animals using the software Mark 8.1 (White & Burnham, 1999). Sex, time and their interaction were included as factors in the models (Lebreton et al., 1992). First, we tested the fit of the full time-dependent Cormack–Jolly–Seber model by groups (model {Phi(g*t) p(g*t)}, where g is sex and t is time) using program Release. The results indicated highly significant lack of fit, due to heterogeneity in males (Test2 +Test3, males χ287 = 204.25, p < 0.001; females χ231 = 8.31, p = 1). We therefore increased the number of groups to account for BW and HW males. This model with three groups was again not appropriate due to heterogeneity in the group of BW males (Test2+Test3, BW males χ275 = 176.59, p < 0.001; HW males χ215 = 2.81, p = 1; females χ231 = 9.72, p = 1). Finally, we subdivided BW males into two behavioural categories (dominant and submissive), and found that the model fitted all the groups, except for dominant BW males, which still showed lack of fit (Test2+Test3, dominant BW males χ243 = 70.85, p = 0.005). Nevertheless, the overall fit was acceptable (Test2+Test3, four groups χ2146 = 145.46, p = 0.497), and we started with this model, given that no other phenotypic variables could be used to subdivide the group of dominant BW males.

Then, we estimated the variance inflation factor (c-hat; needed to correct for overdispersion) using two procedures: by dividing the c-hat obtained from model {Phi(g*t) p(g*t)} by the mean c-hat of the bootstrap simulations in Mark, and by dividing the mean deviance of the saturated model by the mean deviance of the bootstrap simulations. The first method yielded an estimation of c-hat = 1, and the second c-hat = 1.007. We used the second value (more conservative) to adjust parameter estimates and standard errors. In a first step, we ran models without individual covariates, to select the most supported model, as the one that minimizes the quasi-likelihood version of the Akaike’s Information Criterion adjusted for small sample sizes (QAICc). In a second step, body length was included as an individual covariate.

Reproductive behaviour

During the first 10 days of fieldwork, we conducted focal observations to estimate the limits of the territories and the alternative tactics (dominant and submissive; see ‘Results’) that BW territorial males showed. In territorial damselflies, males tend to occupy first high quality areas, and therefore the amount of fighting for territories increases with the number of males (Switzer, 2002). In P. zoe, little is known about which areas offer the best resources, but males concentrate in the areas closer to the water, where most matings are observed. We assume that males choose the best territories first (those close to the water), and territory quality could be estimated as the average number of males observed in a territory over the three days of focal observations. Copulations were timed to the nearest second. We studied 16 territories, whose limits were defined by the flight path that the territorial male used for patrolling (following Romo-Beltrán, Macías-Ordóñez & Córdoba-Aguilar, 2009). Two observers, from 9:30 to 16:00, carried out focal observations. The duration of each focal observation was 10 min per territory. In these observations, we registered the identity of males, BW alternative tactics and reproductive success (number of copulations and copulation duration).

Dominant and submissive roles: plastic or fixed?

We conducted a manipulative experiment to test whether wing spot size is related to the dominant/submissive roles. To this end, we manipulated black and white spots in four treatments: (i) painting over white and black spots without changing their size (control 1), (ii) unmanipulated males (control 2), (iii) erasing the white spot using black painting (EWS), (iv) increasing the black spot and moving the white spot maintaining its size (IBS) (Fig. 1). We used a black Staedtler® permanent Lumocolor pen and a white Edding® 751 paint marker. Each male was followed for three days after manipulation to establish the territorial male subcategory (i.e. dominant or submissive) as described above (Romo-Beltrán, Macías-Ordóñez & Córdoba-Aguilar, 2009). The response variable was change of status, coded as 0 if the male maintained the previous state (dominant or submissive) and as 1 when he changed status or was found far from the territory. This variable was analysed with a GLM with Bernoulli distribution and logit link, entering as explanatory variables the treatment and the initial status. We compared the change of status between day 0 and the following three days, in separate GLMs.

The number of males that copulated once, twice, and so on, followed a negative binomial distribution (Goodness of fit test, χ22 = 0.004, p = 0.998). Therefore, we used a GLM with a negative binomial distribution and a log link function (with the dispersion parameter estimated from the residual deviance) to test the effect of longevity, body length, territory quality, spot size (mean of black and white spots of fore- and hindwings entered separately), asymmetry and male strategy (dominant or submissive) on the number of matings obtained by each BW male.

Air temperature was obtained from a datalogger situated in the shade, which took a measure every 5 min, to consider this variable in further analyses, as it is known to influence copula duration in odonates (Michiels, 1992). We found that copulation duration did not follow a normal distribution, but was normalized using a Box–Cox transformation with lambda = −0.6. Using a GLM with normal errors, we tested the effect of time of start, temperature, asymmetry, body length, territory quality, age and status (dominant/submissive) on copulation duration.

Statistical tests were carried out using xlStat 2018 software (http://www.xlstat.com) and Genstat 18.1 software (GenStat, 2015). Mean values are presented with their standard error and sample size.

Results

Overall, we observed 931 interactions between BW males. On average BW males considered dominant won 11.05 ± 0.77 disputes (mean  ± SE) and lost 1.17 ± 0.15, whereas BW males assigned as submissive won 0.81 ± 0.14 and lost 9.7 ± 0.67 interactions. Figure 2 shows that there were clearly two groups of males (i.e., submissives and dominants) without any intermediate category.

Morphometrics and asymmetry

Over the period of study, we marked 212 males and 111 females. Table 1 shows the number of BW and HW males, females and their recapture rates. The average size (area) of black spots varied between 0.44 and 1.14 cm2, and white spots between 0.08 and 0.40 cm2 (Table 2). The analysis of the size of the spots in young males that were recaptured after 15–37 days, indicates that spot area diminished with age (Fig. S1). The slope of the regression between initial spot size and final spot size was compared to 1, the expectation if the spots do not change over ontogeny. With the exception of the white spot of the fore wings, the slope was significantly lower than 1 (Fig. S1), indicating that their size diminished with age.

Table 1 The number of individuals marked and recaptured in each category, with their body length.

Survival rate was estimated by model averaging over the three most supported models (Table S1). Recapture rate was different for each day and morph. Here we present the estimates obtained from model {Phi(g, length) p(g)} (where g is group and t refers to time; see Table S1), to allow a comparison between groups. BW, black-winged male; HW, hyaline winged male. Mean ± SE.

Variable	BW dominant	BW submissive	HW	Females	
Number marked	54	133	25	111	
Recaptured (%)	44 (81.5)	61 (45.9)	7 (28.0)	22 (19.8)	
Number of recaptures	6.3 ± 0.7	2.8 ± 0.3	1.9 ± 0.4	1.3 ± 0.1	
Body length (cm)	48.9 ± 0.2	48.1 ± 0.2	43.7 ± 0.3	41.8 ± 0.2	
Daily survival rate	0.926 ± 0.019	0.825 ± 0.022	0.917 ± 0.029	0.888 ± 0.038	
Daily recapture rate	0.558 ± 0.024	0.408 ± 0.024	0.197 ± 0.046	0.062 ± 0.015	

Table 2 Descriptive statistics of asymmetry and spot size (cm2) for BW males of Paraphlebia zoe, in function of their mating strategy (D= dominant, S= submissive).

P-value refers to the comparison between dominant and submissive males using an ANOVA.

Variable	Sample size	Minimum	Maximum	Mean	Std. deviation	p-value	
Asymmetry | D	55	0.035	0.722	0.316	0.159	0.257	
Asymmetry | S	113	0.066	1.066	0.353	0.211		
Fore-black right | D	55	0.556	0.901	0.730	0.077	0.974	
Fore-black right | S	113	0.525	1.013	0.730	0.086		
Fore-black left | D	55	0.547	0.909	0.741	0.087	0.859	
Fore-black left | S	113	0.472	1.144	0.744	0.098		
Back-black right | D	55	0.509	0.905	0.685	0.087	0.664	
Back-black right | S	113	0.452	0.926	0.679	0.088		
Back-black left | D	55	0.497	0.878	0.693	0.089	0.562	
Back-black left | S	113	0.436	0.900	0.684	0.090		
Fore-white right | D	55	0.136	0.309	0.215	0.041	0.651	
Fore-white right | S	113	0.135	0.395	0.212	0.042		
Fore-white left | D	55	0.132	0.298	0.216	0.040	0.957	
Fore-white left | S	113	0.097	0.357	0.216	0.044		
Back-white right | D	55	0.086	0.289	0.157	0.037	0.245	
Back-white right | S	113	0.083	0.266	0.151	0.033		
Back-white left | D	55	0.088	0.227	0.153	0.035	0.851	
Back-white left | S	113	0.082	0.270	0.152	0.035		

The mean asymmetry was 0.341 ± 0.015 (N = 168). There were no significant differences between dominant and submissive males in asymmetry or size of the spots (Table 2). There was a significant negative relationship between body length and total asymmetry (Pearson r =  − 0.278, p < 0.0001; Fig. 3).

Figure 3 The relationship between body length and total asymmetry for BW males of P. zoe.

The line represents the regression equation and the ellipse the 95% confidence interval (Pearson r =  − 0.167, p = 0.283).

Estimation of survival rates

Among BW males, dominant males were resighted a mean of six times in contrast to three times for the submissive males. HW males and females were resighted 1–2 times (Table 1). The analysis of daily survival and recapture probabilities using Mark indicated that the most supported model was {Phi(g) p(g*t)}, implying an effect of the group (three types of males and females) on survival and a variable recapture probability due to the effect of the group, the time and their interaction (Table S1). Introducing body length as an individual covariate improved the statistical support for the model (Table S1). We fitted models with common and different intercepts for the covariate, and also a model with a quadratic term of body length, to test for stabilizing selection. Results indicated that the model with a common intercept and without the quadratic term is the most supported (model {Phi(g*length) p(g*t)}-common intercept}, Table S1). The best three models, all included daily variation of recapture rates and the effect of the group, and their interaction. We estimated survival probabilities using the model averaging option of Mark (Table 1, and present the recapture rates of the first model with the term p(g), to allow a comparison between groups.

The effect of the individual covariate, body length, was positive on survival rate in all types of individuals (Fig. 4). Hence, larger individuals had increased survivorship, even among non-territorial HW males and females.

Figure 4 The relationship between body length and survival rate for territorial (BW), non-territorial (HW) and females of Paraphlebia zoe.

The black line represents the estimate and the grey lines the confidence intervals. In all cases, larger individuals survived better.

Reproductive behaviour

BW males (dominant and submissive) were identified and their reproductive activity in terms of number of matings, sperm translocation and copulation duration was registered.

Sperm translocation occurred in precopulatory tandem, and had a duration of 19.3 ± 0.4 (53), with a range from 12 to 25 s. Copulation duration oscillated between 3 and 16 min, with an average of 6.6 ± 0.5 min (N = 49 matings, all BW males), and only occasionally copulation was interrupted by short breaks. Copulation duration was not correlated with any of the variables measured using bi-variate correlations (results not shown; Spearman r; variables tested: asymmetry, body length, quality of territory (number of individuals in the territory), temperature and age). Using a GLM with normal errors (see ‘Methods’), we found that copulation duration was not different between dominant and submissive males, and the only marginally significant variable was temperature, which had a negative effect (estimate = −0.039 ± 0.020, p = 0.066; Table S2).

We analysed lifetime mating success of marked males. This analysis suggests that longevity had a positive effect on mating success (estimate: 0.042 ± 0.011, t158 = 3.90, p < 0.001; Fig. 5). Territory quality also had a positive effect (estimate: 0.232  ± 0.057, t158 = 4.07, p < 0.001; Fig. 6) and BW-submissive strategy had a negative effect compared to BW-dominant strategy (estimate: −1.040 ± 0.225, t158 = −4.62, p < 0.001). The other variables were not significant (see Table S3).

Figure 5 The relationship between longevity and reproductive success (number of matings) for BW males of P. zoe.

The line represents the regression equation and the ellipse the 95% confidence interval (Pearson r = 0.39, p < 0.001).

Figure 6 The relationship between the quality of territory (number of individuals in the territory) and reproductive success (number of matings) for BW males of P. zoe.

The line represents the regression equation and the ellipse the 95% confidence interval (Pearson r = 0.29, p < 0.001).

Dominant and submissive roles: plastic or fixed?

The sample of 79 BW males, whose spots were manipulated (Fig. 1) to test whether spot size (black and white) had an effect on their mating strategy, were observed during three days after manipulation. Comparing day 0 to day 3, the experimental treatments had no significant effects, suggesting that the strategy was not affected by spot size manipulation (Table 3). However, in two comparisons (day 0 to day 1, and day 0 to day 2) the initial status had a negative significant effect, i.e., animals with dominant status were more likely to become submissive over time than the opposite (Table 3).

Discussion

Our results indicate that BW males can behave as dominant or submissive. We found evidence for a positive effect of territory quality (measured as the number of males occupying the area over time) and lifespan on mating success, and that male morph strategy and sex affected the probability of recapture, and larger individuals had higher survival. However, black wing spots asymmetry was not correlated with the reproductive strategy of BW males, and the experimental manipulation of the black and white wing spots did not affect male dominance. We will discuss these findings in relation to what is known in this species and other territorial animals.

BW dominant males were resighted more often than BW submissive males, HW males and females. This is easily explained since BW dominant males are more prone to stay on a territory for long periods, because they are expected to be the best males, the only ones able to defend the territories close to the water (see also Bañuelos Irusta & Araújo, 2007). In many animals, territorial males are usually larger, and defend their territory from other males continuously (e.g., Baker, 1983), a situation commonly found in other territorial damselflies (reviewed by Córdoba-Aguilar & Cordero-Rivera, 2005; Suhonen, Rantala & Honkavaara, 2008) and other taxa (e.g., Mathis, 1991). Territory holders behave aggressively and this is expected to be costly. In fact, experimentally increasing the level of aggressiveness with methoprene (an analog of the juvenile hormone) impaired fat reserves and reduced survival in Calopteryx damselflies (Contreras-Garduño et al., 2011) and reduced survival in Argia (Córdoba-Aguilar & Munguía-Steyer, 2015). Being large and being territorial can be therefore costly, and the expectation is that only males of high quality are able to defend their territory for long periods of time (Tsubaki & Ono, 1987). In our case, larger individuals had higher survival rates not only in the two BW male morph strategies, but also in HW males and females (Fig. 3). This contrasts with previous results on the same population, that only found a positive effect of size on survival in females but not in males (Munguía-Steyer, Córdoba-Aguilar & Romo-Beltrán, 2010). The general trade-off between territory defence and survival, under the assumption that territory holding is costly (Blanckenhorn, 2000), did not hold in our experiment, suggesting again that BW dominant males were of high quality. In other damselfly species, larger males presented higher survivorship (e.g., Heteragrion cooki: Rivas-Torres, et al., 2017; Polythore mutata; Cordero-Rivera et al., unpublished), and similar results were also found in other taxa, such as the mosquito Aedes aegypti (Maciel-De Freitas, et al., 2007). Large size is favoured in adult odonates, but probably not in larvae, contributing to the stasis in body size observed over evolutionary time (Waller & Svensson, 2017).

Table 3 Results of the experiment of manipulation of black and white spot size of BW males of Paraphlebia zoe.

Values are parameter estimates (±SE) after a GLM with Bernoulli distribution and logit link, where the response variate is “change of status”, coded as 1 if changed or 0 if not. Treatments include (i) painting over white and black spots without changing their size (control 1), (ii) unmanipulated control males (control 2), (iii) erasing the white spot using black painting (EWS), (iv) increasing the black spot (IBS) and moving the white spot maintaining its size (reference level for the comparisons). Strategy refers to the initial reproductive strategy of the males, dominant (D) or submissive (S). Spot size manipulation significantly affected only whether males were dominant or submissive, and only for Day 0 versus Day 1 and Day 0 versus Day 2, as indicated by values of p < 0.05 in bold.

	Comparison between days	
Factor	0 to 1	0 to 2	0 to 3	1 to 2	1 to 3	2 to 3	
control 1 (compared to IBS)	−1.07 ± 1.00	−0.67 ± 1.00	−0.51 ± 0.68	−0.10 ± 1.45	−0.13 ± 0.70	−0.06 ± 0.79	
control 2 (compared to IBS)	−0.99 ± 1.00	0.39 ± 0.88	−0.63 ± 0.71	1.59 ± 1.17	0.00 ± 0.71	−1.57 ± 1.17	
EWS (compared to IBS)	−0.39 ± 0.89	0.71 ± 0.86	−0.93 ± 0.74	0.76 ± 1.27	−0.92 ± 0.81	−1.57 ± 1.17	
Strategy (S compared to D)	−1.80 ± 0.83a	−1.69 ± 0.71b	−0.89 ± 0.52	−0.68 ± 0.79	−0.71 ± 0.53	0.02 ± 0.70	
Notes.

a p = 0.030.

b p = 0.01.

We did not find differences in the asymmetry of the black wing spots between BW dominant and BW submissive males. Nonetheless, our results indicated that larger males (independently of their reproductive strategy) were more symmetric. Therefore, asymmetry per se does not seem to affect the ability of males to behave in a dominant way, but perhaps could affect indirectly via the effect of the body size. Furthermore, those males marked when young and recaptured as mature, indicated that spot size diminished with age, a result in agreement with previous findings in other wing-pigmented damselflies (Mnais: Sanmartín-Villar, 2017; Polythore: Sanmartín-Villar & Cordero-Rivera, 2016). Whether these changes are adaptive, as apparently occurs with changes in coloration with age in other taxa (Dreiss & Roulin, 2010; Lopez-Idiaquez et al., 2016) remains to be studied. It seems, therefore, that black wing spots in P. zoe are important for agonistic interactions, and that over their life, males lose part of their signals of quality. The expectation is that old males will be unable to hold territories when confronted by young freshly coloured males (reviewed by Suhonen, Rantala & Honkavaara, 2008).

Our observations on mating behaviour and copula duration are consistent with other studies on the genus Paraphlebia, with the exception that in P. zoe copulation was usually not interrupted, a fact commonly observed in P. quinta, where it was interpreted as evidence for sperm ejection by females (González-Soriano & Córdoba-Aguilar, 2003). We found that copulation is brief for odonates, with an average of 7 min, a value almost identical to previous studies of the same species (Wong-Muñoz et al., 2013), and of similar duration for dominant and submissive males. During our fieldwork period, HW males were very rare, but previous studies showed that HW males copulate for longer (Wong-Muñoz et al., 2013). Figure 5 shows that mating success was positively related to longevity, a result that is general in many odonates (Fincke, Waage & Koenig, 1997; Koenig, 2008) and other animals (e.g., Madsen & Shine, 1992). Given the unpredictable nature of weather, particularly in some regions, animals able to live for long periods are likely to be more successful (Cordero, 1995), and in our species this correlates with a large body size. The fact that previous studies have also found a positive effect of body size on male success in P. zoe (Bello-Bedoy et al., 2015) is an argument for the necessity to demonstrate these effects in more than one study. Contrasting results between studies could be due to variable selection regimes among years, which is a reason to replicate research (Kelly, 2006) and increase our confidence on the generality of results.

Another factor that correlates with mating success is territory quality (Fig. 6). In our case, the territory quality was estimated as the number of males that occupied the area over time, although only one of them was the dominant. Note that we assumed that males are able to identify the best territories (but see below). Moreover, the higher the number of BW submissive males that surrounded a dominant BW male, the more successful the dominant BW male was, suggesting that submissive BW males do not represent a real threat for the dominant BW male, or at least the presence of submissive BW males does not completely counterbalance whatever factors are affecting mating success of the BW dominant male. Submissive BW males probably behave more as sneakers, which would make this system particularly interesting as HW males may also behave as territorials, despite its more common satellite strategy, as found by Romo-Beltrán, Macías-Ordóñez & Córdoba-Aguilar (2009). These authors found that some territories were consistently more attractive for females than others, which is similar to our findings (Fig. 6). They suggested that males did not recognize the best territories because in their study male turnover did not correlate with the number of matings observed in each area (Romo-Beltrán, Macías-Ordóñez & Córdoba-Aguilar, 2009). Nevertheless, this could depend on the season: when density is low, some territories would be vacant, and males could in these cases perform better by occupying a vacant site than by trying to fight for an already occupied territory. Being able to find and defend the best territories, i.e., those that attract a higher number of females, seems to be the key for success in P. zoe males, and territorial defence is apparently related to fat reserves (Romo-Beltrán, Macías-Ordóñez & Córdoba-Aguilar, 2009).

Contrarily to our predictions, we found that manipulating black and white wing spot size did not change behavioural status in BW males; submissive males remained submissive, even after an enlargement of their black spot. In agreement with this, we did not find differences in the black and white wing spot size between dominant and submissive males (Table 2), and the size of the black and white wing spots did not correlate with mating success. In birds, increasing the size of the spots, changed the aggressive behaviour of manipulated males (Senar, 2006). Our reasoning was that wing spot size would be relevant in intra-sexual selection (Romo-Beltrán, Macías-Ordóñez & Córdoba-Aguilar, 2009), but apparently the effect, if any, was negligible. The only change detected was a tendency for dominant males to lose their status over time (Table 3), but without relation to the experiment. It is possible that in species whose males do not court, as P. zoe, females have to rely on cues that are not related with the attractiveness of the males (like wing spots) and follow rules related with the resources offered by the territories, which in this species are basically the best sites to oviposit.

Conclusions

Our findings add further arguments to the recognition of ethodiversity as a basic element of biodiversity (Cordero-Rivera, 2017). We have found that Paraphlebia zoe, a species with male dimorphism (BW and HW morphs) associated with alternative reproductive tactics, holds high behavioural diversity, because at least three types of strategies, dominant, submissive and satellite, are concealed in males. This intra-morph behavioural diversity might be related to male phenotype and environmental factors, such as population density or different proportions of BW:HW males in different periods of the reproductive season (Wong-Muñoz et al., 2013), which may affect male reproductive behaviour in damselflies (Cordero-Rivera & Andrés, 2002). Performing similar manipulative experiments with the HW males when they are in sufficient density would be of great relevance to understand the relationship between male morph and dominance. Furthermore, future studies should take into account the UV wavelength of the black and white bands, which in other species have been found to relate to territoriality (Xu & Fincke, 2015).

Supplemental Information

Table S1 Model selection using QAICc.

The notation for models follows Lebreton et al. (1992) , where g is group (three types of males and females) and t is time. The effect of body length (length) was included as an individual covariate in some models, sometimes as a quadratic effect (length2). Estimates corrected for c-hat = 1.007.

Click here for additional data file.

Table S2 Results of the GLM analysing the effects of phenotypic and environmental variables on copulation duration of P. zoe.

Click here for additional data file.

Table S3 Results of the GLM analysing the effects of phenotypic and environmental variables on lifetime mating success of P. zoe.

Click here for additional data file.

Figure S1 Ontogeny of the black and white-spots areas in BW males of P. zoe. Spot size (in cm2) of ten BW males which were measured at two stages; young and matures (after 15–37 days).

The original size is presented on the horizontal axis, and the final size in the vertical axis. If the spot does not change over ontogeny, the expectation is a line with a slope of 1 (red diagonal). Results indicate that all spots diminish in size, but the tendency is not significant for the white spot of the forewings. The equations for regressions of both sides are presented. The p-value tests the hypothesis of slope = 1.

Click here for additional data file.

Data S1 Raw data

All necessary data to replicate our calculations.

Click here for additional data file.

We are grateful to Pedro Démanos, Janet Nolasco, Daniel Tokman, Fernanda Baena and Ramsés Chávez for field assistance, and to Rogelio Macías Ordóñez for his suggestions and advice during all the study period. Thanks to M. Olalla Lorenzo-Carballa for the help on manuscript editing, and to two anonymous referees, who greatly improved the readability of our text.

Additional Information and Declarations

Competing Interests

Author Contributions

Data Availability

The authors declare there are no competing interests.

Anais Rivas-Torres conceived and designed the experiments, performed the experiments, analyzed the data, prepared figures and/or tables, authored or reviewed drafts of the paper, approved the final draft.

Rosa Ana Sánchez-Guillén conceived and designed the experiments, contributed reagents/materials/analysis tools, authored or reviewed drafts of the paper, approved the final draft.

Adolfo Cordero-Rivera conceived and designed the experiments, analyzed the data, contributed reagents/materials/analysis tools, prepared figures and/or tables, authored or reviewed drafts of the paper, approved the final draft.

The following information was supplied regarding data availability:

The raw data are available in Data S1. The file include tables with the raw data on reproductive success, mark-recapture and manipulative experiments.

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
