# Peer review of "Alternative reproductive strategies in black-winged territorial males of Paraphlebia zoe (Odonata, Thaumatoneuridae)"

_PeerJ, doi:10.7717/peerj.6489_

## Round 0.1 · original submission · Major Revisions

Please revise your manuscript paying particular attention to the comments about experimental design. Note that your paper will be returned to the reviewers and there is no guarantee that it will be accepted after revision, given the seriousness of the criticisms.

Reviewer 1 ·

Basic reporting

The English used is generally clear but should be improved for a general readership. For example, in l. 38 I think the authors probably mean 'In contrast' rather than 'Nevertheless,' the word 'oscillated' is used when 'ranged' or 'varied' would be more idiomatic (e.g. l. 197), and 'associated to' is used but 'associated with' is correct (e.g. l. 39). There are occasional problems with subject-verb agreement, e.g. ll. 226 and 311.

The cited literature gives sufficient context given for the research and is relevant, and the structure of the article is appropriate.

Although the table/figures are appropriate, more information should be provided about some of them in the legends for clarity's sake. For example, in Table 1, giving the result in one-two sentences would be helpful (e.g. 'Spot size manipulation significantly affected only whether males were dominant or submissive, and only for Day 0 versus Day 1 and Day 0 versus Day 2.'). Some figures lack needed information. For example, are the confidence intervals in Fig. 1 95% confidence intervals? Also the word 'males' should be in that figure's legend. Is the X-axis in Figure 2 in days?

Experimental design

The research is within the scope of the journal, and there are some good data in this paper addressing the behaviors of male color morphs. However, the research questions and the author's approach to them are confusing. For example, in ll. 76-80, the authors write 'The aim of our study is to determine whether dominant and submissive roles of BW males are condition-dependent [sic], in particular in relation to asymmetry and size of the wing pigmentation....To investigated [sic] that, we first, using mark-recapture methods, estimated survivorship rates and the effect of body size on survival of BW males (dominant and submissive0. The problem here is that the methods do not address the putative question. It is thus difficult to determine the authors' aims.

In addition, the methods are incomplete. Examples of fundamental questions remaining: Was this research conducted in a stream? A pond? How much area was covered? How often were surveys conducted and at what times of day? Exactly what marks were used to identify individuals, and how did these marks not interfere with the spots/stripes on males' wings?

The research questions are clearly defined individually, but the links among them are not clear, and as noted above the methods do not always match the goals. The authors would have a paper with greater clarity if it were more focused.

Validity of the findings

It is difficult to assess the validity of the findings without more detailed methods, but the analyses are generally appropriate. More than three days of focal observations would have been very helpful and made the findings more convincing.

The authors might explain more fully in the introduction why they predict painting submissive males with larger black spots would change their behavior.

Additional comments

This is an interesting topic, and as you point out there are not many species with these kinds of polymorphisms that have been well studied. If you can re-organize your data, write the paper so that tests of your hypotheses are logical and clear, improve the flow, and clarify the methods, the manuscript would be much improved.

Reviewer 2 ·

Basic reporting

The manuscript is well organized and structured and written in professional English. However, it is confusing in part due to some critical typos and confusion between the types of male damselflies (BW and HW) that undermine a clear description/understanding of the experimental design. The description of the wings of the males is not completely described, ignoring chromatic elements that may be very relevant to their potential as intraspecific signals. In addition there are two studies that explore similar questions about a very similar damselfly that are not considered (Xu & Fincke, 2015; Schultz & Fincke, 2009). They are very relevant to this study. Figures and tables are OK but table S2 should not be in supplementary materials but should be included with the body of the text. Given the importance of conveying what the different morphs look like, photos of the damselflies and wing treatments would be better if they appeared with the text.

Experimental design

This species and the topic of alternative morphologies and reproductive strategies are very interesting and worth studying. The experimental design is not clear due to several critical misprints and conflation of morpho-types (BW and HW) males. There are three critical flaws in the study. First, the alternative personalities of the territorial BW males, categorized as dominant or submissive, are not supported with quantitative measurements of the insects behavior such as number of aggressive pursuits or strikes or retreats as a percentage of opportunities for male-male interactions. There are only descriptions of aggressiveness that could be challenged as subjective. I can see no reason why the so-called submissive males should not be treated as satellites. Secondly, in the exploration of plasticity in the territorial roles of male damselflies by manipulating the wing spots, the authors apparently hypothesized that changing spot size would change the aggressiveness of the male. It is unlikely that the insects have a sense of self, but rather their territorial status is determined by how rival males react to the wing treatments. For example, does reducing the size of the spots of a dominant male result in more or prolonged challenges from other males. Finally, as far as I can tell, the resource holding power of the males are assessed by their persistence in a territory. That is OK when comparing existing physical parameters such as size and asymmetry or outcomes such as reproductive success, but it is a very indirect measure of the effect of intrasexual signals. The actual behaviors of competing males needs to be observed.

Validity of the findings

The data comparing physical attributes and reproductive success of territorial and satellite males are valid and informative. But the manipulative experiments testing the plasticity of the territorial roles and the significance are not valid so the results are inconclusive rather than negative. Perhaps the authors should drop this portion of the study from the manuscript, but that makes it less interesting. In my opinion, they should follow their previous documentation of significant morphological differences between the BW and HW males, and pursue the exploration their alternative strategies and wing patterns.

Additional comments

Introduction

Reviewer's comments: I am perplexed by the description and consideration of the wing spots on BW males. Photographs of the species in the field indicate that the dark portions of the wing tips are iridescent blue in reflected light but may appear dark or black in diffuse or transmitted light. Why is the wavelength of spots ignored given that the color may well be an important component of the visual signal? The wing spots of BW males in P. zoe are remarkably similar to the wing patterns of Megaloprepus coerulatus in which males are also territorial. Why do the authors ignore recent studies (Schultz & Fincke, 2009; Xu & Fincke, 2015) of the function and significance of wing patterns in this latter species which may be informative in understanding P. zoe? Xu & Fincke Xu (2015. Ultraviolet wing signal affects territorial contest outcome in a sexually dimorphic damselfly. Animal Behaviour 101:67-74) showed that territorial status of males was related to the size of the white spots, which may in turn be honest signals of fat reserves since the spots consist of extruded wax. The potential significance of the white spots of the BW males of P. zoe was largely ignored in this manuscript.

Text pp. 65-72 "BW males that do not defend a territory have been previously found in P. zoe (Romo-Beltrán, Macías-Ordóñez & Córdoba-Aguilar, 2009), and are considered as satellite males. Our preliminary observations suggested that BW territorial males can behave as dominants, when their territory is close to the water or as submissive, when they occupy a territory away from the water, in a similar way as the behaviour described for D. obscura (Bañuelos Irusta & Araújo, 2007). Dominant BW males are highly territorial and highly aggressive against submissive (males), probably giving them resource advantages (Baker, 1983), and submissive males are less aggressive, usually avoiding confrontation with dominants, but nevertheless defend their territory against satellites."

Reviewer's comments: In trying to understand how dominant and submissive BW males were identified, I read the cited papers. It is not clear to me how territorial but submissive BW males differ from satellite males as described in Romo-Beltran et al. (2009). The Bañuelos Irusta & Araújo (2007) study distinguished the males as territory holders or satellites, not as two types of territorial males with different degrees of aggressiveness. I cannot find in this manuscript or in the previous studies of P. zoe any quantitative evidence for recognizing distinct categories of “personality” (dominant or submissive) or Resource Holding Power (RHP).

Text lines 95-97: "Based on previous studies with odonates that have found that more pigmented males have higher success (e.g. Alex Córdoba-Aguilar, 2002) we hypothesized that increasing the black-wing spot of the HW submissive males would change their status towards dominance, while increasing the black spot size of the BW males would not affect their role, because they are already dominant."

Reviewer's comments: Previous discussion above (lines 67-69) indicates that dominants and submissives are sub-categories of BW males. The HW submissive males referred to in line 96 must be a misprint and would seem to refer to BW submissives, since HW males don’t have black spots to enlarge.

Text lines 98-102: "For the white spot there is no information about its function. So that, if we conceal the white spot, we could either expect that BW dominant males change their status to submissive if this spot is related to aggressiveness, or that they maintain the same dominance, if the spot is not related to dominance. In the case of submissive BW males, concealing their white spot would not affect their behaviour under the assumption that the white spot is not related to dominance."

Reviewer's comments: The logic here seems flawed. The text repeatedly implies that manipulating the wing spots will change the behavior of the territorial (dominant or submissive) males, when what is relevant is how rival competitors perceive the altered males and lower or escalate their efforts in displacing the territory holder. It appears that a change in status as a territorial holder was determined by whether the altered male was still on the territory. This is only an indirect and imprecise measure of a change in territorial status as a function of the wing manipulation. What is required is direct observation of contests between males and a determination of which male wins.

This is all the more confusing because of the existence of the HW males, some of which are territorial (Romo-Beltran et al. 2009). I would have expected the authors to manipulate the wings of the HW males and record any change in territorial status and RHP versus the BW males.

Methods

Text lines 107-108: "Males and females were captured with an entomological net and marked with a black permanent pen."

Reviewer's comments: Marked how? On wing? If on the wing, how is it known that the markings don’t affect responses of conspecifics.

Text lines 109-113: "Male behavioural status was divided into two categories: territorial males (BW), defined as those individuals who stayed in the same place for more than two days and returned to the same perch site after participating in a fight in flight with a conspecific male, while individuals who did not meet this criterion were considered as non-territorial HW (satellite) males (Romo-Beltrán, Macías-Ordóñez & Córdoba-Aguilar, 2009)."

Reviewer's comments: Again confusing. Are HW males the same as satellite males. Wouldn’t HW males be distinguished by wing morphology rather than behavior? Misprint? BW instead of HW?

116-118 Submissive males were defined as those that perched inside a territory, but further away from the water and at a distance of 0.5-1 m from the dominant male, and were unlikely to engage in fights…..

Reviewer's comments: Why aren’t BW males that perch away from water regarded as satellites rather than submissive territory holders. How do you tell the difference. In Calopteryx, such males would be considered to be satellites since a territory contains a resource to be defended. Returning to the same perch does not necessarily indicate territorial behavior.

And again, how was the likelihood to engage in fights quantified?


Text lines 171-175: "We conducted a manipulative experiment to test if wing spot size is related to the dominant/submissive roles. To this end, we manipulated black and white spots in four treatments: (i) painting over white and black spots without changing their size (control 1), (ii) un-manipulated males (control 2), (iii) erasing the white spot using black painting (EWS), (iv) increasing the black spot and moving the white spot maintaining its size (IBS) (Figure S2)."

Reviewer's comments: The logical first step is to determine if there are differences in the black and white wing spots between BW dominants and BW submissives in the field. The authors apparently did this, but did not describe the procedure in the methods. The results appear in supplementary table S2 cited on line 241. This data is too important to not include with the main body of the text. It is an indirect test of the significance of the wing spots as intrasexual signals.

What media were used to paint over the spots? This is important to report should other researchers wish to conduct a similar experiment.

Results

Reviewer's comments: The metrics comparing the morphology, survival rates, reproductive activity, and lifetime mating success are all solid and informative. The results regarding the plasticity of dominant and recessive roles are flawed for the reasons outlined above. The measurements/analysis of spot size for dominant and submissive BW males should be reported here. The results of continued residency on territories by manipulated males are not a direct measure of RHP.

Discussion

Text lines 249-255: "Our results indicate that: i) BW males can behave as dominant or submissive: we found evidence for a positive effect of territory quality (measured as the number of males occupying the area over time) and lifespan on mating success, and that morph strategy and sex affected the probability of recapture, and larger individuals had higher survival. However, black wing spots asymmetry was not correlated with the reproductive strategy of HW males, and the experimental manipulation of the black wing spots did not affect male dominance. We will discuss these findings in relation to what is known in this species and other territorial animals."

Reviewer's comments: Differences in the behavior of so-called dominant and submissive are not quantified, but the potential reproductive consequences associated with the two putative behaviors is evident from the study. It is perplexing to me why the authors perceive these apparent differences in aggressiveness as alternative reproductive strategies rather than conditional tactics. The reproductive behaviors of the BW and HW males would seem to be alternative strategies and worthy of further pursuit. It would be VERY interesting to see the effect of manipulating the wings of HW males.

---

## Round 0.2 · Minor Revisions

Please excuse my earlier decision; I did not see the data on wins and losses because I expected to see these data in the Results section, and in a regular figure. Please move this information on the number of wins and losses for dominants versus submissives into the Results section and make a regular figure for it (not one in the supplemental section). Make sure you explain how you determined a win versus a loss in the Methods. Also make sure you have a proper figure legend. Thank you and I will be happy to review your paper again once you have made these changes.

· Appeal

Appeal


· · Academic Editor

Reject

Unfortunately when I re-evaluated your revised manuscript I found that many of the reviewers' original problems with it had not been corrected. For example, a major problem is that the paper contains no data showing that "dominant" males actually won more aggressive contests. I am sorry I cannot give you better news, but hope the reviews can help improve your paper for another journal.

#

---

## Round 0.3 · Major Revisions

I am sorry, but your revisions do not adequately address my request. Saying that dominant males won "most" of the interactions and subordinate males lost "most" of the interactions is completely inadequate. You need to give a more precise definition. For example, did you determine a male as dominant if he won 60%, 75%, 90% of his interactions? What about subordinates? And you need to put this more precise definition in the figure legend as well as the Methods.
Finally, there are lots of corrections to the writing that I have marked directly on the manuscript (attached). Please fix these before returning it,

---

## Round 0.4 · accepted · Accept

Thank you for making these final revisions to your manuscript.

#